# Immune Response after Vaccination against Tick-Borne Encephalitis Virus (TBEV) in Horses

**DOI:** 10.3390/vaccines12091074

**Published:** 2024-09-19

**Authors:** Dana Kälin, Angela Becsek, Helen Stürmer, Claudia Bachofen, Denise Siegrist, Hulda R. Jonsdottir, Angelika Schoster

**Affiliations:** 1Clinic for Equine Internal Medicine, Vetsuisse Faculty, University of Zurich, 8057 Zurich, Switzerland; angela.becsek@vetmeduni.ac.at (A.B.); helen.stuermer@uzh.ch (H.S.); angelika.schoster@lmu.de (A.S.); 2Equine Internal Medicine, University Equine Clinic, University of Veterinary Medicine Vienna, 1210 Vienna, Austria; 3Institute for Virology and Immunology, Department of Infectious Diseases and Pathobiology, Vetsuisse Faculty, University of Bern, 3012 Bern, Switzerland; 4Spiez Laboratory, Federal Office for Civil Protection, 3177 Spiez, Switzerland; 5Department of Rheumatology, Immunology, and Allergology, Inselspital University Hospital, 3010 Bern, Switzerland; 6Department of BioMedical Research, University of Bern, 3012 Bern, Switzerland; 7Equine Clinic, Centre for Clinical Veterinary Medicine, Faculty of Veterinary Medicine, LMU Munich, 80539 Munich, Germany

**Keywords:** TBEV, TBEV vaccination, equine, IgG, IgM, neutralising antibodies, immune response, SAA

## Abstract

(1) Background: Horses infected by a tick-borne encephalitis virus (TBEV) can develop clinically apparent infections. In humans, vaccination is the most effective preventive measure, while a vaccine is not available for horses. The objective of this study was to describe the immune response in horses after a TBEV vaccination with a human vaccine. (2) Materials and Methods: Seven healthy horses were randomised to a treatment or a control group in a stratified fashion based on TBEV–IgG concentrations on day −4. The treatment group (*n* = 4) was intramuscularly vaccinated using an inactivated human TBEV vaccine on days 0 and 28; the control group (*n* = 3) did not receive an injection. A clinical examination and blood sampling were performed on day –4, 0, 2, 4, 6, 8, 10, 14, 28, 30, 32, 34, 36, 38, 43, 56, 84, and 373. A linear mixed model analysis was used to compare IgG and IgM concentrations, neutralising antibody (nAb) titres, leucocyte count, serum amyloid A (SAA), and fibrinogen and globulin concentrations between the groups and time points. (3) Results: The clinical examination was normal in all horses at all time points. There were no significant changes in SAA, globulin, and fibrinogen concentrations and leucocyte count between the groups or time points (all *p* > 0.05). There was no significant increase in IgG, IgM, or nAb titres in the control group over time (all *p* > 0.05). In the vaccination group, there was a significant increase in IgG concentration and nAb titres after the second vaccination (*p* < 0.0001). There was no significant increase in IgM antibodies after the TBEV vaccination (all *p* > 0.05). One horse in the vaccination group had an IgM concentration above the laboratory reference on day 10. (4) Conclusions: The human TBEV vaccine did not have side effects when used in healthy horses in this study. A significant rise in TBEV-specific IgG antibodies and nAbs after the second vaccination was observed. However, IgG and nAb titres have been shown to decrease within 1 year after vaccination. The results of this study indicate that a vaccination with a human vaccine only induces a mild rise in IgM antibodies and only in previously naive horses. With no significant changes to inflammatory parameters in the vaccinated horses, it remains unclear whether vaccination with the human vaccine leads to protective immunity.

## 1. Introduction

Tick-borne encephalitis (TBE), caused by the flavivirus tick-borne encephalitis virus (TBEV), is currently the most important viral tick-borne zoonosis in Europe [1]. The TBEV is presently endemic in parts of Europe and Asia. The TBEV is transmitted by Ixodes Ricinus ticks and small mammals, especially rodents, that act as reservoirs. Large wild mammals, birds, domestic animals including horses, and humans can be infected and develop the disease but do not contribute to the spread of the TBEV due to low and short viraemia [2]. While most TBEV infections are subclinical or associated with mild general illness, rare cases of encephalitis have been reported in humans, dogs, monkeys, deer, sheep, goats, and horses [3,4]. Clinical signs in affected horses are usually mild but can include fever, lethargy, anorexia, behavioural changes, reduced consciousness, stupor, ataxia, recumbency, muscle fasciculations, epileptic fits, shivering, paralysis, and proprioceptive deficits [4,5].

TBEV seroprevalence in horses has been investigated in various studies in different countries in Europe and ranged from 2.9 to 37.5% [6,7,8], depending on geographical location, age structure, and management practises of the study population. The seroprevalence of the TBEV in the equine population in Switzerland has not been investigated to date. TBEV infections in horses can be diagnosed by serology, virus isolation, or RT–PCR. The latter two techniques are limited to the early viraemic phase of the disease, as at the onset of neurological signs, the virus is cleared from the blood [9]. Furthermore, RNA load is low in horses during the viraemic window [10]. Diagnosis is therefore primarily based on detection of antibodies against the TBEV. A two-step method is recommended. Due to the cross-reactivity of different flaviviruses in enzyme-linked immunosorbent assays (ELISAs), positive or questionable ELISA results should be confirmed by a serum neutralisation test (SNT) [4]. Neutralising antibodies (nAbs) are also the best marker for a protective humoral immune response induced by vaccination [9]. In general, little is known about antibody production in infected horses. Neutralising antibodies have been detected in one horse on day 21 post experimental infection [5,11] and at the time of the first hospitalisation in seven naturally infected horses—all of these horses were also positive in an IgM ELISA [5,12]. TBEV-specific IgG has been described to persist for at least 9 months in horses [4] but is believed to persist for many years.

In humans, vaccination is the most effective preventive measure against TBE. In Europe, two vaccines are accredited for use in humans, namely FSME-Immun (Baxter Vaccine AG, Vienna, Austria) and Encepur (Chiron-Behring, Marburg, Germany). Both products are whole-virion vaccines based on a formaldehyde-inactivated Neudoerfl strain and the German isolate K23, respectively [1]. Both vaccines have been demonstrated to be highly effective; 75.3–83.5% of vaccinees seroconvert after a single dose and up to 100% after two doses [13,14]. Additionally, no serious adverse vaccine-related events have been reported so far [1]. Currently, there is no licenced vaccine available for veterinary use. Nevertheless, FSME-Immun has been experimentally tested in different animals in various studies. After the vaccination of dogs, cattle, sheep, goats, rabbits, pigs, mice, and one horse, most immunised animals developed TBEV-specific antibody titres. The vaccinated horse did not show any clinical signs of illness nor allergic reactions and TBEV-specific antibodies could already be detected one week after the first immunisation [15].

No study to date has investigated IgM or inflammatory marker responses to the TBEV vaccination in horses. This study aims to further investigate the use of the human vaccine in horses by vaccinating a small study population and describing the production of IgG and IgM antibodies and changes in clinical parameters and inflammatory markers.

## 2. Materials and Methods

### 2.1. Study Population

A prospective experimental trial was conducted in northeastern Switzerland between January 2022 and January 2023. The main study period was chosen in the low-tick season to minimise the chance of natural infection during the basic immunisation, as this region is a known TBEV risk area. The study population consisted of seven healthy mares aged between 5 and 21 years (mean 14 years, Table 1) from a research and teaching herd from the University of Zurich. The mares were housed at the same location for at least 1 year before the study and throughout the study. The mares were group-housed in a barn with permanent access to an outdoor sand paddock and daily access to a nearby pasture. They were fed ad libitum hay and a balancer once daily and had access to water at all times. The management was not changed as part of the study.

### 2.2. Experimental Design

An overview of the study design is illustrated in Figure 1. On day −4 of the study, 6 mL of blood was taken from the jugular vein of all seven mares using the BD Vacutainer System with 6 mL serum tubes (BD Vacuette), and TBEV-specific IgG antibodies in the serum were measured using a commercial ELISA (Immunozym FSME IgG all-species ELISA, Progen, Heidelberg, Germany) according to the manufacturer’s instructions. The mares were then assigned to the treatment group (V-1 to V-4) and control group (C-1 to C-3) in an alternating fashion based on decreasing IgG concentrations, with the mare with the highest IgG concentration being assigned to the control group (Table A1).

At the beginning of the study (day 0), all seven mares were clinically examined. Clinical examination included general condition, behaviour, posture, the presence and quality of faeces and urination, respiratory rate, heart rate, rectal temperature, thorax and gastrointestinal auscultation, the appearance of mucosal membranes, the presence of nasal or ocular discharge, body surface temperature, the filling of jugular veins, skin turgor, the intensity of facial artery pulse and digital pulses, the appearance of the injection site (left neck), and the presence of ticks. Blood was sampled from all seven mares for haematology, biochemistry, and serology using the BD Vacutainer System. A 20 G BD Vacutainer needle was inserted into the jugular vein and one K3 EDTA sample tube (6 mL), one lithium heparin tube (6 mL), one serum tube (6 mL), and one 9NC coagulation sodium citrate 3.2% tube (2 mL) were filled (all sample tubes were from BD Vacuette).

A complete blood count (Sysmex XN-1000 (Sysmex AG, Kobe, Japan)) and serum biochemistry (Cobas 6000 c501 (Roche Diagnostics International AG, Rotkreuz, Switzerland)) were performed within two hours of collection at the central laboratory of the Vetsuisse Faculty of the University of Zurich, Switzerland. Reference intervals (RIs) for inflammatory parameters measured on these two machines were as followed: leucocyte count: 4.7–8.2 × 10^3^/µL; SAA: 0.5–1.2 mg/dL; and globulin concentration: 1.3–2.9 g/L. Fibrinogen concentration (RI:1.3–2.9 g/L) was measured on a Start Max (Diagnostica Stago S.A.S., Asnière sur Seine, France). Blood samples for serology were centrifuged at 2000 rpm for eleven minutes within 2 h after sampling and serum was separated into 1.5 mL Eppendorf tubes and stored at −80 °C until analysis.

All horses were considered healthy based on both clinical and laboratory exam findings. The horses in the treatment group were administered 0.5 mL of the human vaccine FSME-IMMUN CC (Pfitzer AG, Zurich, Switzerland) containing 2.4 µg of the inactivated TBEV strain Neudoerfl by intramuscular injection into the left neck on day 0 and day 28. The three mares in the control group did not receive an injection. The clinical examinations and blood sampling were repeated as described on days 2, 4, 6, 8, 10, 14, 28, 30, 32, 34, 36, 38, 43, 56, 84, and 373. Complete blood count and biochemistry were performed on days 2, 4, 28, 30, 32, and 84. On day 373, only six horses were available for sampling.

### 2.3. Serological Testing

Frozen sera were thawed for measurements of IgG, nAb, and IgM concentrations. IgG measurements were performed at the Institute of Virology at the Vetsuisse Faculty of the University of Zurich, Switzerland. The Immunozym FSME IgG all-species ELISA kit (Progen GmbH, Heidelberg, Germany) was used according to the manufacturer’s instructions. The reference for this kit as per the manufacturer was as followed: <63 seronegative; 63–126 borderline; and >126 seropositive.

To determine nAb titres against the TBEV, aliquots of frozen samples from selected days (Figure 1) were sent to Spiez Laboratory, Spiez, Switzerland, for a SNT. A TBEV SNT was performed as previously described [16,17]. Briefly, sera were heat-inactivated for 30 min at 56 °C a day before analysis [18] and stored at +4 °C. On the day of analysis, samples were diluted 1:8 in Leibovitz L-15 medium (Biochrom AG, Berlin, Germany) supplemented with 5% foetal bovine serum (FBS, Seraglob, Schaffhausen, Switzerland). Thereafter, fivefold dilutions were made in duplicates in a 96-well plate (TPP Techno Plastic Products, Trasadingen, Switzerland). Under biosafety level 3 conditions, 100 TCID50 TBEV (Hypr, provided by Daniel Růžek, University of South Bohemia, České Budějovice, Czech Republic) were added and incubated overnight at 4 °C, followed by incubation at 37 °C for 1 h without CO_2_. Subsequently, 15,000 porcine kidney stable (PS) cells provided by Daniel Růžek, University of South Bohemia, České Budějovice, Czech Republic, were added to each well, and the plates were incubated at 37 °C without CO_2_. On day 3, neutral red dye (final concentration of 0.000165%) in Dulbecco’s phosphate-buffered saline (Sigma Aldrich, Buchs SG, Switzerland) was added to each well. On day 5, the presence or absence of neutral red-stained cells was used to assess the virus-induced cytopathic effect (CPE). Alternatively, plates were stained with 80 µL of crystal violet on day 5 for CPE assessment. Neutralising capacity was reported as the geometric mean titre (GMT) of two replicates. In all neutralisation tests, the neutralisation titre was defined as the reciprocal dilution resulting in at least a 50% virus neutralisation. The lower limit of detection (LLOD) was a titre of 1:16.

For IgM measurements, aliquots of the frozen serum samples were sent to Laboklin GmbH & Co. KG, Bad Kissingen, Germany, where they were analysed using the VetLine TBE/FSME IgM ELISA kit (NovaTec Immundiagnostica GmbH, Dietzenbach, Germany) according to the manufacturer´s guidelines. The kit is a “research use only” test kit and it has not been validated by the manufacturer to date, so there are no official reference intervals and test characteristics known. Laboklin GmbH & Co. adapted the manufacturer´s recommended reference intervals for use in animal sera as follows: <25 LE seronegative and >25 LE seropositive (LE = Laboklin–Einheit is calculated as follows: LE = (Sample absorbance value × 10): cut-off sample absorbance value.)

### 2.4. Statistical Analysis

The statistical analysis software (SAS) R (version 4.2.1) was used for the statistical analysis of IgG and IgM concentrations and inflammatory parameters. A linear mixed model was used to compare the IgG ELISA concentration, nAb titres, leucocyte count, and SAA, fibrinogen, and globulin concentrations between the two groups as well as between different time points. IgM concentrations were analysed using a generalised linear model with a gamma link. A *p*-value of 0.05 was considered statistically significant. For the remainder of the data, descriptive statistics were used.

## 3. Results

### 3.1. Clinical Findings

No abnormal clinically relevant findings were detected in the clinical examinations of any of the horses throughout the study period except a systolic grade 4/6 and a diastolic grade 2/6 heart murmur in horse V-1, which was a previously known condition. No ticks were found on the horses throughout the study. There were no signs of local inflammation detected at the injection sites and no horse showed a rise in rectal temperature after vaccination.

### 3.2. IgG Concentrations

IgG concentrations over time for each horse are shown in Figure 2, group means over time are shown in Table 1, and individual values can be found in Table A2 (Appendix A). On day 0, five out of seven horses were IgG-negative based on the manufacturer’s cut-offs (VIEU/mL; reference range: <63 negative, 63–126 borderline, >126 positive). The initially seronegative horses in the vaccinated group seroconverted during the study period with an at least fourfold increase in antibody concentration, one horse after the first vaccination, and two after the second vaccination (Figure 2 and Table A2). The initially seropositive horse in the vaccination group also displayed a threefold rise in antibodies after the second vaccination (Figure 2 and Table A2). Two out of the three horses in the vaccination group that were available for sampling on day 373 were still seropositive but showed a two to threefold reduction in IgG concentrations compared to day 84 (Table A2). In the control group, the initially seropositive horse (C-2) remained seropositive at several sampling time points without an obvious rise or fall in antibodies (236.42 +/− 100.05 VIEU/mL, Figure 2 and Table A2). One of the initially seronegative horses (C-3) had a seropositive result on days 2 and 4 and a threefold rise in antibodies on day 32 which was sustained until day 84. This horse still had a seropositive IgG titre on day 373 that had only decreased 1.8-fold compared to day 84 (Figure 2 and Table A2).

Over the course of the study, there was a significant increase in IgG concentrations only in the vaccinated group (group × time interaction *p* < 0.001); there was no significant effect of group or time alone on the IgG concentration (*p* > 0.05; for *p*-values, see Table 1).

### 3.3. Neutralising Antibody Titres (nAb)

Serum neutralisation tests (SNTs) were performed to confirm that the antibodies detected via the ELISA were indeed directed against the TBEV and not against other flaviviruses. Selected time points were analysed.

Concentrations of nAbs over time for each horse are shown in Figure 3a,b, group means over time are shown in Table 1, and individual values can be found in Table A2. All horses in the vaccination group were negative on day 0 and seroconverted on day 10, three horses stayed seropositive from then on, one horse was negative for nAbs by day 28. A 10-fold increase in titres was seen after the second vaccination on day 28 in all horses. On day 373, the three horses available for sampling were still seropositive, with one horse maintaining high concentrations of nAbs. In the control group, two horses (C-1 and C-2) were seropositive on day 0 and remained positive at various sampling timepoints; however, they were without a significant rise in nAbs. The horse in the control group that showed a significant and sustained rise in IgG antibodies in the ELISA from day 32 onwards (C-3) did not display a similar sustained rise in the nAb titre. It only had a marginally positive result at one single time point of the study (day 38) but was seronegative again on day 84.

The vaccinated group had significantly higher nAb titres than the control group, while time alone did not have a significant effect on this parameter. The increase in nAb titres over time in the vaccinated group compared to the control group was significant (for *p*-values, see Table 1).

### 3.4. IgM Concentration

IgM concentrations over time for each horse are shown in Figure 4, group means over time are shown in Table 1, and individual values can be found in Table A2. On day 0, all horses tested negative (reference: <25 LE) for TBEV IgM antibodies (Figure 4, Table A2). Only one horse in the vaccination group tested positive during the study (day 10: 25.53 LE). Time, group, and time × group interactions had no significant effect on IgM antibody concentrations (all *p* > 0.05; for *p*-values, see Table 1).

### 3.5. Inflammatory Parameters

Inflammatory parameters of each horse are shown in Table A3 and group means are presented in Table 1.

One horse in the vaccinated group had leucocytosis at all sampling points throughout the whole study period. In the control group, two horses developed leucocytosis on different days of the study. There was no significant effect of group, time, or group × time interactions on leucocyte count (for *p*-values, see Table 1).

Every horse had an elevated SAA concentration at some point during the study; however, as most values remained <20 mg/dL, these marginal elevations were considered insignificant [19]. One horse in the control group displayed significantly elevated SAA (452.0, 462.3, and 158.9 mg/dL) on days 28, 30, and 32, respectively, preceding the rise in IgG antibodies in the ELISA. There was no significant effect of group, time, or group × time interactions on SAA concentrations (see Table 1 for *p*-values).

Hyperfibrinogenaemia was detected in horse C-3 (3.5 g/L) on day 28 and in horse V-1 (3.0 g/L) on day 84 of the study. There was no significant effect of group, time, or group × time interactions on fibrinogen concentrations (for *p*-values, see Table 1).

All globulin concentrations remained within the reference range throughout the study period and there was no significant effect of group, time, or group × time interactions on globulin concentrations (for *p*-values, see Table 1).

## 4. Discussion

This study showed that horses mount an immune response to vaccination with a human TBEV vaccine without showing signs of adverse reactions or changes in inflammatory parameters.

No adverse reactions to the vaccination were noted in the four mares vaccinated over the course of this study. In one of the other published studies in which a horse had been vaccinated with the same vaccine, the horse did not show any side effects either [15]. In humans, adverse reactions are usually mild. Depending on the study, 0.25–2.6% of vaccinees develop pyrexia which lasts for 24–48 h [13,14,20,21]. Fever that lasts ≤ 24 h has been reported after vaccination against Equine herpes virus 1 and 4 (EHV-1/4) in horses [22]. Pyrexia in any of the mares vaccinated in this study could have potentially been missed, as the mares were not clinically examined until 48 h after vaccination. When the same human vaccine was tested in other animals in previous studies (mice, sheep, goats, cattle, dogs, rabbits, and pigs), no adverse reactions or pyrexia were observed in the vaccinated animals [15,23,24]. However, the study population size of the current study is too small to claim that the use of the human vaccine is safe in horses.

All four vaccinated horses in this study mounted an IgG response. Two out of four mares seroconverted after the first vaccination; one mare seroconverted after the second injection. Additionally, the mare that was seropositive from the start showed a significant increase in IgG concentrations after the second vaccination. Similar seroconversion rates have been found in studies in humans, where 75.3–83.5% of vaccinees seroconverted between 21 and 28 days after the first vaccination, with seroconversion rates reaching 100% 14 days after the second vaccination [13,14]. The 100% seroconversion rates after three vaccinations with the same vaccine have been found in pre-clinical trials in mice too [25]. More studies with larger study populations will have to be conducted to calculate more exact seroconversion rates after the vaccination of horses against the TBEV. Similar IgG titres were reached in the vaccinated horses in this study 28 days after the first injection (mean 261.19 VIEU/mL) compared to in a study in humans (260–340 VIEU/mL) [13]. Another study in humans using the same vaccine as in this study found similar IgG concentrations 21–35 days after the second vaccination (mean 631.3 VIEU/mL) [20] as in our study in horses on day 56 (mean 576.31 VIEU/mL). It can therefore be concluded that IgG concentrations in horses rise to similar concentrations after vaccination against the TBEV as they do in humans. However, in all the studies conducted in humans, the study population was naive before the first vaccination, which was not the case for the study in horses. Seroconversion rates and IgG concentrations are therefore not directly comparable. Nevertheless, the horse with the lowest IgG titre at the beginning of the study (V-1) developed IgG concentrations comparable to the ones found in previously naive humans after the first and second vaccination. Two of the three horses that were retested one year after the second vaccination remained seropositive, which was consistent with expectations, as IgG is known to persist for several years [1,10]. A study in humans found similar results, with a large proportion of the vaccinees remaining seropositive 300 days after the first vaccination, though with a decrease in antibody titres [13], which was the case with the horses in the current study as well. A study in children found a slow but steady decrease in IgG in both the ELISA and SNT over the course of five years after three vaccinations [26]. Similar results have also been found in goats and sheep and in one horse after the TBEV vaccination [24,27]. This decline in IgG concentration within 1 year after vaccination indicates that regular booster vaccinations will be necessary in order to maintain immunity. Studies in humans have shown significant increases in antibody concentrations after a booster vaccination [21,26], so it can be assumed that this would be the case in horses too. However, this has not been investigated to date. It is interesting to note the variance between the individual study horses at the time points at which the horses seroconverted after vaccination, the level of IgG concentrations, and the persistence of antibodies after one year. Individually different serological responses have been found after the vaccination of humans too [13,14,26]. In the only other study where a horse was vaccinated against the TBEV, antibody concentrations were measured with a modified version of the Immunozym FSME IgM kit to determine total immunoglobulin concentrations (IgG plus IgM) after vaccination [15]. The results of that study are therefore not directly comparable with the results of our study. The horse in the study by Klaus et al. was vaccinated with the same vaccine as used in this study but using a double dose at weeks 0, 1, 3, and 11 and showed an increase in antibody titres one week after the first vaccination followed by a short temporary decrease in weeks 5 and 6 and an increase again after the fourth immunisation. In the same study, mice, dogs, cattle, sheep, goats, rabbits, and pigs were immunised with the same vaccine and vaccination scheme as the horse. Except for some individual mice, all of the vaccinated animals seroconverted. The majority seroconverted after the third vaccination and a smaller proportion only after the fourth immunisation. A rise in antibodies after the first vaccination—like in the vaccinated horse in that same study—was only observed in rabbits. This rise in antibodies one week after the first vaccination is similar to the increase in IgG in two of the four vaccinated mares in the current study. However, in our study, no significant temporary decrease at week 5 or 6 in any of the horses was detected. It might be that the initial rise in antibodies detected by Klaus et al. after the first vaccination in the horse and the rabbits was in fact an IgM response rather than an IgG response, and only the second rise was due to rising IgG, which could explain the temporary decrease that was lacking in our study horses. It is worth stating again that there were large differences in time points of seroconversion and maximum titres reached between the individual animals, similar to the individual responses of the horses in the current study. The horse in Klaus et al.´s study was still seropositive 21 weeks after the first vaccination, which is in line with results of the current study. In goats and sheep, a similar decline in antibody titres over the course of several months but with still positive titres after 28 months has been found [27], as in our study horses after one year.

IgG concentrations in the vaccinated horses reached concentrations similar to what has been described by Fouché et al. in a naturally infected horse, which had an IgG concentration of 381.7 VIEU/mL at the onset of CNS signs, rising to >500 VIEU/mL four weeks later [5]. Klaus et al. have compared IgG titres of a horse with a suspected previous natural TBEV infection with a vaccinated horse and found that titres were comparable after four immunisations [4]. It could be concluded that vaccination therefore leads to high enough antibody concentrations for protective immunity. High nAb titres in the vaccinated study horses support this hypothesis, as nAbs are meant to be the best markers for protection [9,28]. All vaccinated horses seroconverted in the SNT after the first vaccination; however, a significant increase in nAb titres was only seen after the second vaccination, which is comparable to the IgG ELISA results. Similar high nAb titres have been found in mice after three vaccinations with the same vaccine (1:983 ± 126) [25] as in the horses in the current study after two vaccinations (1:611.56 ± 187.39). In both the SNT and ELISA, all retested horses showed a decrease in antibody titres 1 year after the TBEV vaccination. As the SNT is considered the gold standard and most specific diagnostic test for detecting TBEV-specific antibodies [4,7], these results are probably more accurate than the ELISA results. In this study, the SNT seems to be more sensitive than the ELISA as well, detecting seroconversion earlier than the ELISA in some of the study horses. However, we used a multispecies ELISA for IgG determination that uses a single cut-off for all species. Further studies may be necessary to adjust this cut-off specifically to horse sera.

It is unclear why one horse in the control group (horse C-3) seroconverted in the IgG ELISA on day 14 of the study, reaching nearly as high IgG concentrations as the horses in the vaccinated group after the second vaccination. Interestingly, in the same horse, there was a significant increase in fibrinogen concentration on day 28 and in SAA concentration from days 28 to 32, reflecting an acute inflammatory response to an unknown stimulus, but no increase in TBEV IgM. An increase in SAA and fibrinogen concentration alone is very unspecific; however, in this case, the concurrent rise in TBEV-specific IgG supports the diagnosis of a recent infection with a flavivirus. One explanation could be an infection with a different flavivirus to the TBEV, like the West Nile virus (WNV) or Usutu virus, as there is known cross-reactivity between flaviviruses in the ELISA [7]. Negative SNT results in this study horse—except one mildly seropositive value on day 38—support this hypothesis, as the SNT has the highest degree of specificity of all diagnostic tests [10]. TBEV IgG ELISA test results could therefore be a false positive in this horse due to cross-reactivity with a different flavivirus. However, no other flavivirus is known to be endemic in the area where the study was conducted. Furthermore, the immunisation started in January and therefore, a mosquito-borne infection with WNV or another flavivirus is very unlikely within the first 2–3 months of the experiment. Another explanation for seroconversion in this horse could be a natural infection with the TBEV by a tick bite. It was an unusually mild March in the year when the study was conducted, which could have led to premature tick activity. Negative SNT results in this horse seem to make it less likely though that IgG was in fact directed against the TBEV. However, the results of the conducted SNT might not be accurate and could be a false negative. It therefore remains unclear whether IgG found in this horse is directed against the TBEV or against a different flavivirus.

There was no significant increase in IgM concentration after vaccination of the four mares in this study. Only one horse (V-1) developed IgM antibodies, and they were only detectable on a single occasion 10 days after the first vaccination. This horse was the youngest out of the study population and had a very low IgG titre at the beginning of the study, indicating that this horse had not been exposed to the TBEV before the start of the study. A study in humans found that only a small proportion of vaccinees (3/9) develop detectable concentrations of IgM after vaccination against the TBEV and titres are generally low and of short duration (up to 240 days), with the steepest rise in IgM within four weeks after the first injection [29]. A different study found an even lower IgM seroconversion rate in humans (8/150 vaccinees) [9]. In the study by Klaus et al., where a horse was vaccinated against the TBEV, the ELISA to measure antibody response did not distinguish between IgM and IgG, so it is unknown whether this horse developed IgM or not [15]. After vaccination of previously naive horses against the WNV, another flavivirus, IgM was detected in 32% of samples taken after the first vaccination [30] and in 10% of samples taken after the second vaccination or after an annual booster vaccination [31]. In conclusion, all these studies in humans and horses show proportions similar to the one found in the current study and reflect that IgM is more likely to rise after the first vaccination, which was the case in the one horse that seroconverted in the current study as well. It might also explain why, in this study, the horse with the lowest IgG titre at the start of the study was the one to develop IgM antibodies. However, it is unclear why, in this particular horse, an IgM peak was only measured at one time point (day 10) and declined so rapidly—within 4 days—after that. It could have been a false positive result. The fact that the other three horses in the vaccinated group had likely been exposed to the TBEV before the start of the study could explain why none of them mounted an IgM response. In contrast to low proportions of detectable IgM concentrations after vaccination, almost all horses developed IgM upon a natural TBEV infection. Six out of seven horses with clinical TBE in Switzerland had IgM above the cut-off concentration at the onset of clinical signs [5,12]. The only horse that was retested 32 days later was seronegative at this point [5], reflecting the early rise and rapid decline typically found with IgM with natural infections [32]. In humans, IgM was also discovered in almost all patients with clinical TBE [9]. It is important to note that the IgM ELISA kit used to analyse the study samples is a “research use only” kit and has not been validated by the manufacturer to date, meaning that test characteristics like precision, diagnostic sensitivity and specificity, interferences, and the cross-reactivity of the test kit are unknown. In addition, it is an “all-species” ELISA kit and species-specific reference intervals have not been defined to date. Therefore, IgM results in this study have to be interpreted with caution. The lack of an IgM response in three of the four vaccinated horses could thus also be explained by a potentially low sensitivity of the test used or by a cut-off that might have been set too high for this species and subsequent false negative test results. Likely a combination of all of the above reasons has led to the detection of low IgM concentrations in the study horses. In addition, the study population size in this study is too small to claim that horses do not mount IgM antibodies after vaccination with the human TBEV vaccine. Studies with larger study populations will be necessary for any valid conclusions about IgM production after the vaccination of horses.

No significant changes in leucocyte count, SAA, fibrinogen, or globulin concentrations were found in the horses in the vaccinated group compared to the control group. Although the youngest horse of the study population (V-1, 5 years old) did in fact show leucocytosis throughout the study period, this is unlikely to be connected to the vaccination, as the leucocyte count was already high prior to the first vaccination. No cause for this sustained leucocytosis could be found in clinical examinations of this horse nor for the temporary leucocytosis in two mares in the control group. In contrast, various studies have found changes in inflammatory parameters after the vaccination of horses with inactivated vaccines. Horses vaccinated against Equine influenza and tetanus showed a significant increase in SAA and fibrinogen concentrations and leucocyte count [33]; horses vaccinated against EHV-1/4 showed an increase in SAA concentration [22,34]; and an increase in SAA and γ-globulin concentrations was also found in horses after vaccination against WNV [19,35]. In all of these studies, changes in inflammatory markers were most significant 24 to 48 h after vaccination but were still detectable three days after vaccination, and some lasted even longer. As the horses in the current study were not sampled before 48 h after the first and second vaccination, changes earlier than that could have potentially been missed. However, one would still expect to detect changes at 48 h. One explanation why the horses in this study did not show an inflammatory response could be the relatively high mean age of the study population of 14 years, as it has been described previously that older horses are less likely to respond to vaccination with an increase in SAA concentration [19]. Another possible explanation for the lack of inflammatory response in the study horses might be the fact that an inactivated vaccine made for use in humans was used. In inactivated vaccines, it is usually the adjuvants—not the antigen itself—that stimulate the native immune system, and it may be that the adjuvants in the human vaccine are not potent enough to cause an inflammatory response in horses, especially when considering that the dose of the vaccine was not adjusted to the horses´ bodyweights in this study; instead, the same dose as for vaccination of a human was used. As acute inflammatory responses are thought to be essential for vaccinations to initiate immunity [22], it is unclear whether a rise in IgG without an inflammatory response will be enough to stimulate immunity against the TBEV. Therefore, the type and/or dose of adjuvants might have to be altered in the human vaccine for use in horses. However, the aim of the inactivated vaccine is not to develop lifelong defence against the TBEV, but rather to create memory lymphocytes to help prevent severe clinical and fatal infections, which was probably successful in the vaccinated mares. Further studies with larger and younger naive study populations and potentially with an altered vaccine are needed to investigate if the vaccination of horses with the human vaccine induces immunity against infection with the TBEV, and not only an increase in IgG, as this alone cannot be taken as proof of protective immunity.

## 5. Conclusions

The human TBEV vaccine did not have side effects when used in healthy horses in this study. However, the study population size was too small to be able to conclude that it is safe to be used in horses. A significant rise in TBEV-specific IgG antibodies and nAbs after vaccination was observed. However, both IgG and nAb titres have been shown to decrease within 1 year after vaccination, which will make regular booster vaccinations necessary. The results of this study indicate that vaccination only induces a mild rise in IgM antibodies and only in previously naive horses. With no significant changes to inflammatory parameters in the vaccinated horses, it remains unclear whether vaccination with the human vaccine leads to protective immunity against infection with the TBEV and the development of clinical disease. However, it was not the aim of the study to prove that vaccination leads to immunity against TBEV infection. Instead, this study aimed to investigate the effects of the human vaccine on clinical parameters, antibody concentrations, and inflammatory markers in a small number of horses as a pilot study. No conclusions should be drawn about the protective effect of the vaccination in horses. Not only was it a small study population, but also the vaccinated horses were not challenged with an infection. Further research with larger, naive study populations and potentially with an altered vaccine, as well as virus challenge trials, will have to be performed before the safe and effective use of the TBEV vaccination in horses can be promoted.

## Figures and Tables

**Figure 1 vaccines-12-01074-f001:**
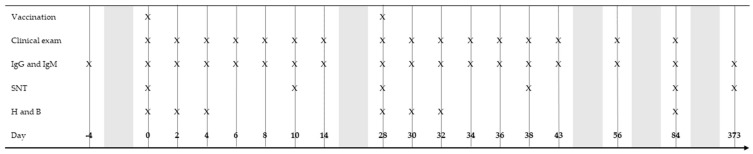
Overview of the study design of an experimental trial investigating the immune response to TBEV vaccination in horses. Days on which the horses were vaccinated (vaccinated group only, *n* = 4) and clinically examined (all study horses, *n* = 7), serum was sampled for IgG and an IgM ELISA and SNT (*n* = 7), and blood was taken for haematology and biochemistry (H and B, *n* = 7), which are indicated with an “X” in the respective row.

**Figure 2 vaccines-12-01074-f002:**
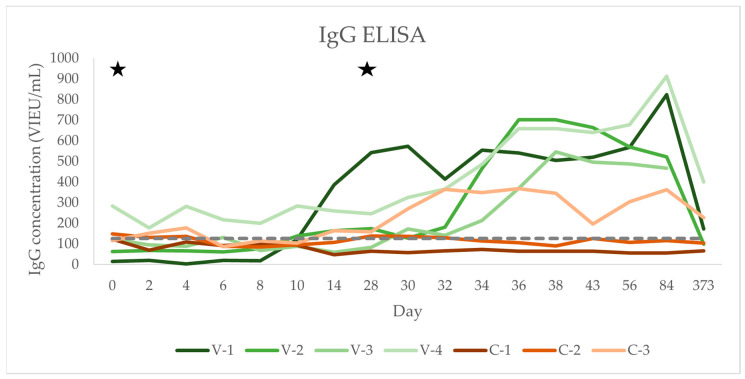
Longitudinal serum TBEV IgG concentration (VIEU/mL) after TBEV vaccination in 4 horses compared to 3 unvaccinated controls. IgG concentrations of horses in the vaccinated group (*n* = 4) are represented by green lines. IgG concentrations of the horses in the control group (*n* = 3) are represented by red lines. Time points of vaccinations are marked by stars. Values above the grey dotted line are seropositive (RI as per ELISA manufacturer: <63 seronegative, 63–126 borderline, >126 seropositive).

**Figure 3 vaccines-12-01074-f003:**
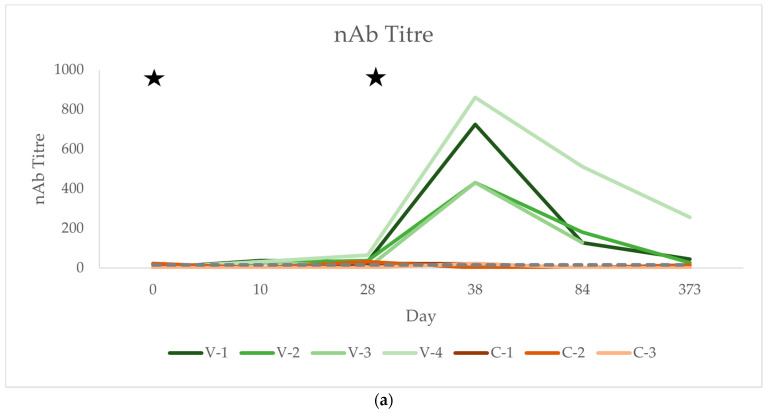
(**a**) Longitudinal serum nAb titres determined using SNTs after vaccination in four horses compared to three unvaccinated controls. Horses in the vaccinated group (V1–V4) are represented by green lines and horses in the control group (C1–C3) by red lines. Time points of vaccination are indicated with stars. Values above the grey dotted line are seropositive (LLOD: 1:16). (**b**) Longitudinal serum nAb titres after TBEV vaccination in four horses compared to three unvaccinated controls, values <50 only. Horses in the vaccinated group (V1–V4) are represented by green lines and horses in the control group (C1-C3) by red lines. Time points of vaccination are indicated with stars. Values above the grey dotted line are seropositive (LLOD: 1:16).

**Figure 4 vaccines-12-01074-f004:**
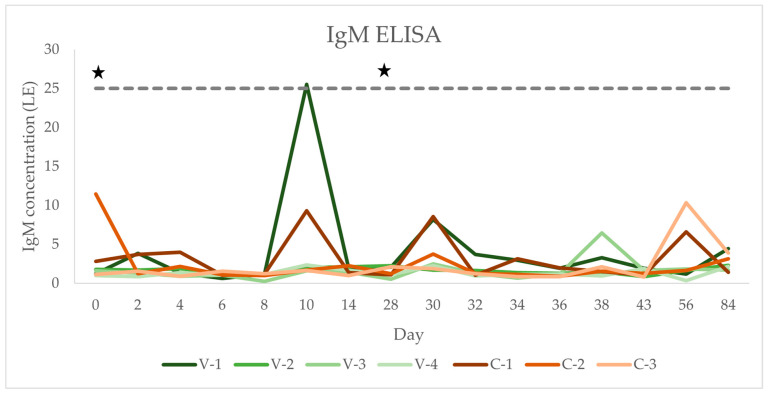
Longitudinal serum TBEV IgM concentration (LE) after TBEV vaccination in four horses compared to three unvaccinated controls. Horses in the vaccinated group (*n* = 4) are represented by green lines and horses in the control group (*n* = 3) by red lines. Time points of vaccination are indicated with stars. Values above the grey dotted line are seropositive (RI as per laboratory: <25 = seronegative; >25 = seropositive).

**Table 1 vaccines-12-01074-t001:** Longitudinal mean (+/−SD) serum TBEV IgG concentrations, nAb titres, IgM concentrations, leucocyte counts, and SAA, fibrinogen, and globulin concentrations after TBEV vaccination in four horses compared to three unvaccinated controls with *p*-values. IgG concentrations (VIEU/mL, RI: <63 seronegative, 63–126 borderline, >126 seropositive), nAb titres (LLOD: 1:16), IgM concentrations (LE, RI as per laboratory: <25 seronegative; >25 seropositive), leucocyte count (RI: 4.7–8.2 × 10^3^/µL), SAA (RI: 0.5–1.2 mg/dL), fibrinogen (RI: 1.3–2.9 g/L), and globulin concentrations (RI: 1.3–2.9 g/L). Horses in the vaccine group (V1–V4) were vaccinated on day 0 and day 28. *p*-values < 0.05 are marked in bold.

	Day	0	2	4	6	8	10	14	28	30	32	34	36	38	43	56	84	373	p Group	p Time	p Group × Time
IgG	Mean Vacc.	120.96	90.49	110.52	107.19	90.52	158.11	217.48	261.19	299.76	275.97	429.63	534.75	602.75	580.47	576.31	681.28	224.04	0.58	0.37	**<0.001**
	SD Vacc.	101.44	57.35	103.81	74.58	66.60	74.65	120.69	172.28	173.72	116.68	129.05	99.85	79.78	73.18	67.76	190.21	128.10			
	Mean Control	129.80	117.90	141.51	89.29	100.97	97.77	106.26	120.37	155.00	186.38	178.62	179.42	167.00	174.24	156.37	178.15	133.04			
	SD Control	13.90	35.36	28.38	2.01	11.90	4.62	47.25	40.40	88.16	128.24	121.29	134.74	127.09	114.09	107.03	132.60	68.59			
nAb	Mean Vacc.	0.00					32.24		33.51					611.56			237.25	109.39	**0.01**	0.06	**<0.001**
	SD Vacc.	0.00					3.95		22.78					187.39			160.09	103.94			
	Mean Control	14.26					0.00		18.21					14.21			0.00	5.33			
	SD Control	10.14					0.00		13.43					10.10			0.00	7.54			
IgM	Mean Vacc.	1.42	1.93	1.37	1.00	0.94	7.89	1.76	1.51	3.66	1.90	1.54	1.39	3.05	1.58	1.26	2.65		0.69	0.55	0.85
	SD Vacc.	0.31	1.15	0.36	0.24	0.40	10.19	0.26	0.70	2.63	1.07	0.87	0.31	2.16	0.44	0.57	1.06				
	Mean Control	5.13	2.17	2.36	1.24	1.15	4.23	1.55	1.49	4.74	1.18	1.72	1.24	1.73	1.05	6.21	2.83				
	SD Control	4.54	1.10	1.26	0.24	0.11	3.61	0.54	0.46	2.82	0.11	1.02	0.51	0.26	0.19	3.57	1.03				
Leucocyte count	Mean Vacc.	7.33	6.95	6.75	6.70				6.50	7.55	7.57						7.88		0.89	0.23	0.61
	SD Vacc.	2.11	1.82	2.27	1.99				1.71	1.96	1.31						2.21				
	Mean Control	7.17	7.23	6.53	6.57				5.80	6.90	6.53						7.67				
	SD Control	1.27	1.52	0.88	1.04				0.79	1.31	1.17						1.39				
SAA	Mean Vac	0.65	1.60	1.75	2.63				0.00	0.78	1.10						1.45		0.38	0.58	0.67
	SD Vacc.	0.82	2.23	1.79	1.57				0.00	1.34	0.34						2.51				
	Mean Control	1.70	3.93	6.27	1.87				150.67	155.27	52.97						0.00				
	SD Control	2.13	5.01	7.49	1.03				213.07	217.11	74.91						0.00				
Fibrinogen	Mean Vacc.	1.68	1.70	1.73	1.85				1.98	1.78	1.90						2.13		0.44	0.31	0.71
	SD Vacc.	0.19	0.25	0.18	0.38				0.30	0.33	0.37						0.53				
	Mean Control	1.83	1.90	1.77	1.90				2.40	2.07	2.20						2.03				
	SD Control	0.17	0.28	0.21	0.28				0.78	0.59	0.50						0.33				
Globulin	Mean Vacc.	31.00	29.50	32.50					28.50	29.00	27.50						32.25		0.34	0.69	0.55
	SD Vacc.	2.24	1.12	0.87					1.80	1.22	1.12						1.48				
	Mean Control	31.33	32.33	32.67					29.33	29.00	29.67						32.00				
	SD Control	2.05	1.25	1.70					0.94	3.56	2.05						2.83				

## Data Availability

The original contributions presented in the study are included in the article/Appendix A; further inquiries can be directed to the corresponding author.

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
