# Peer review of "Immune Response after Vaccination against Tick-Borne Encephalitis Virus (TBEV) in Horses"

_vaccines, 2024, doi:10.3390/vaccines12091074_

Round 1

Reviewer 1 Report

Comments and Suggestions for Authors

The number of horses in the treatment group and the number of horses in the control group is limited. Since n = 4 for the treatment group and n = 3 for the control group, I am wondering if the authors can repeat once more to see if the results can be reproducible that n = 3 or 4 for the treatment group and the control group. Since the limitation of samples, the authors cannot conclude that “ The human TBEV vaccine did not have side effects when used in healthy horses in this 449 study, so its use in horses can be assumed to be safe.”

The main question that the authors would like to know is how the human TBEV vaccine will behave in healthy horses. The authors investigated the immune responses in response to the human TBEV vaccine in horses.

It is good that the authors think that trying the human TBEV vaccine in the horse to investigate the immune response. From the research article itself description, this is a research gap for Switzerland.

In addition to the horses, since the authors used the human TBEV vaccine, it definitely has a lot of preclinical tests and clinical trials tests that have been published. Can the authors compare the current data tested in horses to this vaccine tested in mice and other clinical trials?

The experiment needs to be repeated at least one more time to come up with the solid data and then we can discuss the conclusion.

The references can be improved after the manuscript is modified since the authors need to add a lot of new information to the manuscript.
• Additional comments on the tables and figures.

What I felt was that table 1, and table 2 can be put into supplementary data, if possible, to make the paper more organized.

The authors need to provide explanations to explain why the IgM cannot be detected in immunized horses. In the human body, most vaccine immunizations start at IgM and then go to IgG. Horses could be different from humans but needs a clear explanation.

The figure quality is poor. Try not to simply copy and paste from excel, at least removing gray lines from the excel files.

Author Response

Thank you for reviewing our work. Please see the attachment for our responses to your comments.

Reviewer 2 Report

Comments and Suggestions for Authors

Kalin et al., describe an animal experiment, when 4 horses were immunized with inactivated human vaccine from the market, and several branches of their immune reactions were compared to similar data of 3 control horses. English, tables, figures are good. A virus challenge was really missing to make a whole circle from this work. The authors would not risk too much with such a challenge. The authors got the result one should expect, the vaccine worked as in humans (we are also a mammal species) with unfortunate results of an inactivated vaccine i.e. relatively low IgG level, and short decrease of VN titers, making booster vaccine administrations necessary.  
This and similar works hopefully will pave the way for usage of inactivated TBEV vaccine(s) in animal, veterinary practice soon.

Keywords: equine and horse. Both are necessary? Synonimes.

lines 52.53. Were reported. But clinical signs in an mammal species is extremely rare. All of these reports (dog, sheep, horse) were single case reports, not epidemic, and not about endemic regions for these species. The authors should indicate here, that clinical signs in these species are very rare and only isolated cases were observed. Additionally the described clinical signs are almost always mild.
line 57 Ref? 2,9-37,5%. And be careful with drawing general conclusion from some simple diagnostic studies.

Mat met.
lines 145-146 Intracutan administration of the vaccine would mimick better the natural TBEV infections.
line 158 – why was heat inactivation necessary? It was already done by the vaccine manufacturer.
lines 157-174 – A reference would be needed about this VN test, if exist at all. Or a working VN test. I am doubtful about TBEV VN tests, I have never seen a really, simply working TBEV VN test like in case of adeno and herpesviruses, where such VN tests are reliable clearly working tests. TBEV does not provoke visible CP, unfortunately.

Results
The initially seropositive horse should have been omitted from the experiment, only seronegative animals should have been used, or the seropositive horse should be member in the vaccinated group. Its results confuse the outcome.
Table 2. Even this can be figure out from the numbers, but buti t would help if the vaccinated and control groups would be indicated somehow. It would make the Table more understandable.
line 256. This horse is the same individual with high lasting IgG level (lines 211-212)? Should be indicated.
Figure 2 and 3a. if I understand properly, the initially seropositive horse in the control group had detectable IgG but no neutralizating Abodies? Reason?

Discussion
The authors should not forget, that aim of an inactivated vaccine is not to develop lifelong defence against a certain microorganism, but to create lifelong memory lymphocites, which help to avoid severe clinical signs, and fatal infections in future of the vaccinated individual (Chinese human coronavirus SARS vaccine). And for this respect probably the vaccination described in this paper was successful. We should not expect more. The authors should indicate this in this chapter.

In result lines 198-212 the authors (in limited number of individuals only) detected the signs of individually different serological response to TBEV antigens. We experience almost the same during human vaccinations. This should be mentioned in the discussion chapter.
line 291 – Why showed control horses leucocytosis? Any reasons? wound, infections?

376-386 – If IgG ELISA was false with samples of this horse, why were not false with other horses? + When are the peak activities of larvae and nymphs, in this region of Switzerland? The horse could be infected by TBEV e.g. from a larva. SNT. As I know, a reliable SNT is missing in Flavivirus (TBEV) diagnosis. We use various type of tests, because we eagerly need a Flavivirus species-specific serological test. We apply them, play with them, but do not trust them. All TBEV SNTs are very subjectives, not objectives. And when a diagnostic test is even a little bit subjective and not reproducable, than it is not good. Discussing the results of this horse do not be misleaded by result of a TBEV SNT.

lines 387 – 417 – IgM. As I know IgM antibodies appear first, and decline first. Its detection is used to detect that a TBEV infection is in its early phase or not. Secondly TBEV-specific IgM appears in natural infections but no tor very rarely during vaccination. + This unexpected single IgM positivity could only be a false positive diagnostic result.

lines 444-447 – A virus challenge would be necessary after such a vaccination attempt. This would answer the main questions. But on the base of this study an inactivated vaccine could serve as effcetive in animals like in humans to suppress clinical diseases, and fatal cases.

Comments on the Quality of English Language

spelling
The ends of lines in the first 3 pages are not correct. (e.g. 110, 118)
line 135 Switzerland))
line 152 – mea-surements. But best to avoid any divisions of words. Write in full, the program arrange them in one row.
line 331 – in case of this study.

Author Response

(The authors gave the same response as above.)

Round 2

Reviewer 1 Report

Comments and Suggestions for Authors

The main question that the authors would like to know how the human TBEV vaccine will behave in healthy horses. The authors investigated the immune responses in response to human TBEV vaccine to horses. It is good that the authors think that trying the human TBEV vaccine in the horse to investigate the immune response. From the research article itself description, this is a research gap for Switzerland.

Suggestions and concerns

1.        since the authors used the human TBEV vaccine, it definitely has a lot of preclinical tests and clinical trials tests that have been published. Can the authors compare the current data tested in horses to this vaccine tested in mice and other clinical trials?

2.        The experiment needs to be repeated at least one more time to come the solid data.

3.        The authors need to provide explanations to explain why the IgM cannot be detected in immunized horses. In human body, most vaccine immunizations start at IgM and then go to IgG. Horses could be different from human but needs a clear explanation.

Author Response

Thank you for your time and effort in reviewing this manuscript. Please find our responses to your suggestions in the attached file.

Reviewer 2 Report

Comments and Suggestions for Authors

I have read the authors's response, I accept all of their answers, and I suggest this (TBEV vaccination  in horse) paper in this present form for publication in Vaccine.

Author Response

Thank you for your time and effort in reviewing this manuscript.

Round 3

Reviewer 1 Report

Comments and Suggestions for Authors

I think that this revised manuscript meets the publication requirement.